# Demystify Painting with RL

**Linning Xu**
Department of Information Engineering
The Chinese University of Hong Kong
Shatin, New Territory, Hong Kong
linningxu@ie.cuhk.edu.hk

**Yunxiang Zhang**
Department of Information Engineering
The Chinese University of Hong Kong
Shatin, New Territory, Hong Kong
yunxiang.zhang@ie.cuhk.edu.hk

## Abstract

Given an image of an arbitrary scene, experienced artists are skillful at accurately perceiving the visual contents within the scene, such as objects, lighting and tint, and presenting them in different painting styles. Essentially, this artistic creation procedure starts from a blank canvas and proceeds in a stroke-by-stroke manner, which could be modeled as a sequence of carefully-chosen stroke actions. Given the fact that reinforcement learning (RL) offers a principled approach to tackling sequential decision making tasks, it is natural to consider applying appropriate RL techniques to training machines to mimic the painting procedure accomplished by human artists. In this project, we aim to learn an agent capable of automatically planning a sequence of strokes that result in a painting with desired visual contents and artistic styles, just as what human painters would do during artistic creation. We perform extensive experiments under different algorithmic designs as an attempt to demystify the learning mechanism and capability of current RL-based painting agents. The corresponding video presentation can be accessed using the following link[1].

## 1 Introduction

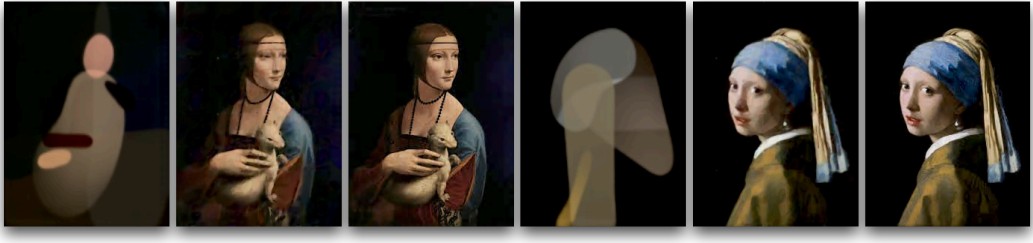

Figure 1: Test results of RL agent on oil paintings *Lady with an Ermine (1489)* and *Girl with a Pearl Earring (1665)*. For each triptych, the target image is shown on the right. The left ones illustrate how our RL agent learns to sketch the outlines, and the middle ones illustrate the final paintings rendered through a sequence of automatically generated brush strokes.

While experienced human artists are skillful at rendering extremely complex scenes into aesthetically pleasing paintings using common drawing tools, it is practically intractable to explicitly formalize the working mechanism underlying the creation procedure, which makes the ability of painting highly difficult to acquire. Within this context, training intelligent machines to imitate the painting process of human artists has been gaining more and more research interest from the computer vision community.

---

[1] https://youtu.be/Nlp4XDcntdk

Given the sequential nature of painting, it is very intuitive to formalize the problem of painting within the framework of reinforcement learning (RL). Specifically, an agent will be trained to automatically generate a sequence of diverse brush strokes such that the resulting canvas satisfies several predefined requirements, such as the resemblance to some target scenes and artistic styles.

In this project, we target the problem of learning intelligent painting machines and depart from the model-based RL framework proposed in *Learn2Paint* [11]. Particularly, we conducted comprehensive experiments under different algorithmic designs to demystify the learning mechanism and capacity of existing RL-based approach to training painting agents, which has not yet been thoroughly studied in the literature. Detailed analysis on different design choices for each component of the overall learning framework will be presented in the sequel, illustrating whether it is possible for the RL painting agent to exhibit controllable behaviors in more diverse scenarios. We conclude with a brief discussion on the limitations of current methods and possible future directions.

## 2   Related Work

The ability of synthesizing non-photorealistic imagery has long been an important research topic in the computer vision and graphics community. Some researchers approach the problem by focusing on images with simpler structural compositions, such as sketches and doodles, while others experiment on images containing richer textures and more complex structures, such as natural images and artistic paintings. In this project, we direct our attention to the study of artistic paintings.

### 2.1   Stroke-Based Rendering

As a representative approach to synthesizing non-photorealistic images, stroke-based rendering (SBR) recreates images by stacking a variety of discrete brush strokes on painting canvas, such as flat wash, hatching, scumbling and stippling [10]. Specifically, the selection of appropriate strokes and their positions at each time step are two key aspects of this approach. Most traditional SBR algorithms address these two aspects either by greedy search on every single step, optimization over an energy function using heuristics [25], or user interaction. Haeberli et al. [9] propose a semi-automatic method which requires the user to set parameters to control the shape of the strokes and select the positions for each stroke. Litwinowicz et al. [16] propose a single-layer painter-like rendering which places the brush strokes on a grid in the image plane, with randomly perturbed positions. Recent deep-learning-based solutions make use of recurrent neural networks (RNNs) for stroke decomposition. Sketch-RNN [7] leverages datasets with sequential information to achieve good results in sketch drawings, but their data collection process requires heavy human labor and it is impractical to extend their approach to more complex images. StrokeNet [27] combines differentiable renderer and recurrent neural networks (RNNs) to train agents to paint but fails to generalize to color images.

### 2.2   Reinforcement Learning for SBR

Recent methods adopt deep RL to learn an efficient stroke decomposition. The authors of SPIRAL [5] propose an adversarially trained deep RL agent that learns the structural information in images, but fails to recover fine details in human portraits. Neural Painters [19] presents a concept called intrinsic style transfer that directly optimize brushstrokes to minimize style transfer's content loss. In this work we will give a detailed investigation on model-based RL framework proposed in Learn2Paint [11], which has shown effective in generating strokes sequentially to recover the target image.

## 3   Neural Stroke Renderer

Traditional rendering pipelines typically involve a discrete operation called rasterization, which makes the rendering non-differentiable. Most earlier works learn the painting policy rely on interacting with such undifferentiable graphic engines which are unable to provide detailed feedback about the generated images. Figure 2 gives an example of such agent. At each time step, SPIRAL [5] outputs program fragments which are rendered into an image via a graphics engine $R$. In this way, the agent can only make use of these intermediate renders to adjust its policy, which prevents it from accessing the direct supervision signal from fine-grained details on the canvas.

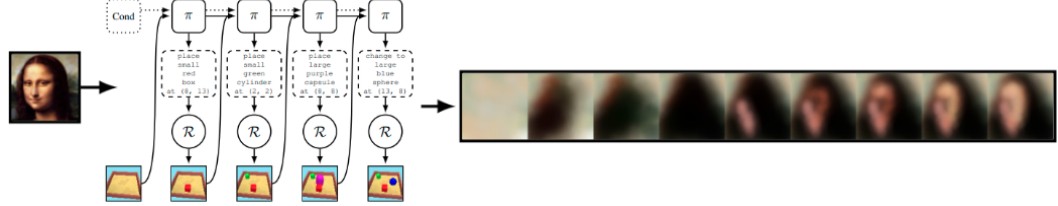

Figure 2: SPIRAL-generated *Mona Lisa*. With a separate graphic engine to render generated images, SPIRAL fails to generate high quality painting with similar granularity as the target image.

Recently, [11; 28] resort to use neural network to approximate the rendering process, serving as the differentiable Neural Stroke Renderer. Besides its merits on enabling end-to-end training, it also provides the flexibility to generate arbitrary styles of strokes. We firstly trained a neural render with Bezier curve same as in [11], and then we provide a more versatile way to incorporate any stroke images which we believe can provide more human-like painting experiences.

**Bezier curve.** Learn2Paint adopts quadratic Bezier curve (QBC) as stroke representation to simulate the effects of brushes. The shape of the Bezier curve is specified by the coordinates of three control points. Formally, the stroke is defined as the following tuple,

$$a_t = (x_0, y_0, x_1, y_1, x_2, y_2, r_0, t_0, r_1, t_1, R, G, B)_t \in R^{13} \tag{1}$$

where $(x_i, y_i)$ is the coordinates of the $i$-th control points of the QBC, and $(r_0, t_0), (r_1, t_1)$ control the thickness and transparency of the two endpoints of the curve respectively. Then the QBC is represented by the area of

$$B(t) = (1 - t)^2 P_0 + 2(1 - t)t P_1 + t^2 P_2, \quad 0 \le t \le 1 \tag{2}$$

In implementation, we divides $t \in [0, 1]$ to $N = 100$ equally spaced discrete values, and generate the stroke from head to tail by repeatedly computing the above QBC equation. We then train our Neural Stroke Renderer (a 6-layer NN) to approximate the above mapping, $f : \mathcal{R}^{13} \to \mathcal{R}^{3 \times 128 \times 128}$.

**Oil Painting Brushes** While Bezier curve has the potential to generate strokes with any shape, this is not practical in real-world paintings. Human artists are usually equipped with only few brushes, yet they can master the strength to control the size, rotation, and opacity of the stroke using a single brush. Also, the stroke generated by Bezier Curve sometimes display very unnatural traces which are impossible to be produced by a human artists. In general, human artists know how to choose brush from a library of brushes that can best fit to current canvas (See Figure 3 for a set of selected brushes). Motivated by this consideration, we re-design the action space as,

$$a_t = (I_{brush}, x, y, \theta, fx, fy, R, G, B)_t \in R^{10} \tag{3}$$

where the random variable $I_{brush}$ is set to select a brush from a collection of brush images (illustrated in Figure 3) and use other parameters to control the specific stroke shape generated by that brush. Specifically, we split $[0, 1]$ into $n$ uniform intervals to match the number of brushes and enforce the action space to maintain continuous. Figure 4 shows a comparison between the generated brush using Bezier curve and oil painting brushes. A recently published work [28] try to shade the texture by blending a texture map on top of all the rendered stroke and simple treat the texture as a constant map which is not updated during the training, which cannot generate enough variation of textures.

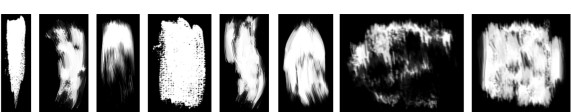

Figure 3: Randomly selected 8 oil painting brushes from internet. The brush images can be replaced with any other brushes that you like: watercolor brushes, pencil, pen, etc.

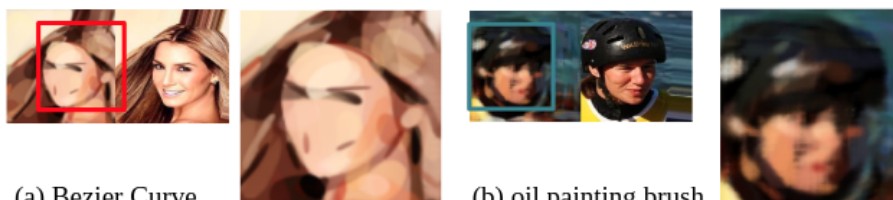

Figure 4: Comparison between rendered image with (a) Bezier Curve and (b) oil-painting like strokes trained with pre-loaded stroke images.

**Supervised Learning**    The neural renderer can be quickly trained with supervised learning and runs on the GPU by computing the $L^1$ or $L^2$ losses. Since the renderer takes in action parameters and direct outputs the whole canvas where a large amount areas are empty, we find that, in order to train with fine strokes, adding a mask on the non-empty parts with a weighted penalty can greatly accelerate the training and enforce it to learn with small-size strokes. We will explain in later sessions that how fine-grained strokes are critical to the success of imitating the target images.

**Model-based RL**    The model-based approach adopted in [11] has proved improved sample efficiency and convergence stability with the design of the neural renderer. While promising, we should be aware that the success of the RL agent training largely depends on the quality of the off-line trained neural renderer. Firstly, in scenarios where the environments are highly stochastic and complicated, especially for those require accurate geometry and lighting conditions may pose huge challenge to the neural render. Secondly, the action space of the agent has to be continuous to efficiently use backpropagation gradient. These restrictions guide us to train the neural renderer for enough long times with the hope to cover as large range of different actions to be learned during the training.

## 4    Reinforcement Learning Agent

While there are many ways to formulate the stroke sequence generation process, here we adopt RL to learn the policies, instead of directly treating it as a parameter optimization like the one taken by [28], which enforces the final rendered output $h_T$ similar to a reference image $\hat{h}$ via $h_T = f_{t=1 \sim T}(\tilde{x}) \approx \hat{h}$, where $x_t$ is a set of stroke parameters. We believe that by learning with RL, we can grasp more insights about human painting process with an artificial painting agent. Furthermore, the exploration and randomness injected in RL can generate more diverse sequences compared with direct optimization formulation.

**Problem Formulation**    Once the stroke renderer is trained good enough, we begin our RL scheme to train the painting agent. Given a target image $I$ and an empty canvas $C_0$, the agent aims to find a stroke sequence $(a_0, a_1, \ldots, a_{n-1})$, where rendering at on $C_t$ can get $C_{t+1}$. After rendering these strokes in sequence, we get the final painting $C_n$, which should be visually similar to $I$ as much as possible. We model this task as a Markov Decision Process with a state space $S$, an action space $A$, a transition function $\text{trans}(s_t, a_t)$ and a reward function $r(s_t, a_t)$. Note that, the transition $\text{trans}(s_t, a_t)$ between states is implemented by the neural renderer, which is known apriori. Thus the training scheme fall into Model-based RL.

### 4.1    Study on State Representation

Learn2Paint [11] separates a state into three parts: states of the canvas, the target image, and the step number. Formally, $s_t = (C_t, I, T)$, where the step number $T$ acts as additional information to instruct the agent the remaining number of steps. Note that, $T$ here is smartly calculated as $T = \frac{t}{\text{Number of max steps}} \in [0, 1]$. While the original work does not explain much about this design, we find that the careful design of state representation and the inclusion of time-stamp indicator is of great significance to the high quality generation.

**Influence of Timestamp**  The state is represented as a 9 channel tensor of $(C_t, I_t, T, P)$ in their official code implementation. [2]. The unmentioned constant variable $P \in R^{2 \times H \times W}$ is the coordinates of each pixel in the canvas, which is a popularly used trick in pixel-wise image generation by teaching the network to be aware of coordinates [15]. Instead, the inclusion of time stamp $T$ is not a natural design, as we may conjecture that the agent should learn to paint a stroke by purely observing the current canvas and the target image, which is independent of time-stamp and can perfectly match with the Markov property. Therefore, we experiments with the setting by taking the $T$ out and repeat with different random seeds. Surprisingly, in many trials the agents are not able to paint with fine-grained strokes and can only grasp a rough outline of the target image. Still, it can success in certain trails and slowly converge to low value losses, yet the learning curve is much unstable than that with $T$ included, and takes a much longer time to get improved (See Figure 5 for comparison).

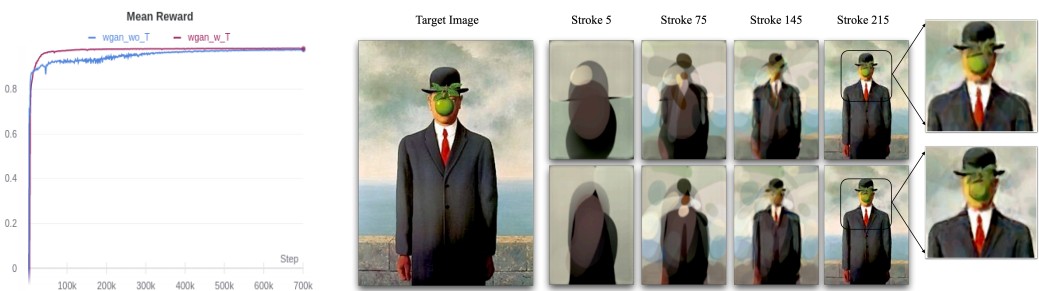

Figure 5: *Left:* Comparison of learning curves of state representation w/o timestamp included. *Right:* Qualitative results comparison step-wise, the 1st line is trained with $s_t = (C_t, I, P)$ and the 2nd line is trained with $s_t = (C_t, I, T, P)$. A more fluently and reasonably generated stroke sequences and a higher quality detailed image is generated with $T$ included.

Additionally, we find that the number of *max steps* can cause quite different performances. By intuition, a longer generated sequence may bring more detailed strokes on the later stage, where the agent can continue improving the finer parts. However, since we take the relative portion to represent the timestamp, a longer allowed max steps do not necessary lead to finer strokes. Instead, we find that the agent will get "lazier" when we increase the max step from 40 to 100, tending to bring smaller reward gains at each time step while maintaining the accumulative rewards approximately the same.

**Hypothesis on Affordances**  The above observation naturally cast the question that "What causes this disparity in sample efficiency w/o timestamp indicator?" In [13], the authors discuss about the theory of affordances in RL, where the term "Affordance" was firstly coined by Gibson [6] in 1977 to describe the fact that certain states enable an agent to do certain actions. [13] established the assumption that on the one hand, affordances in RL allow faster planning, by reducing the number of actions available in any given situation; On the other hand, affordances facilitate more efficient and precise learning of transition models from data. In terms of painting, both by intuition and human experiences, an intelligent agent should start with large strokes to establish the background and the main tone of the whole painting, and then gradually narrow down to focus on fine-grained small strokes. Such kind of affordances established between the timestamp and stroke parameters is likely to explain the above observations.

### 4.2  Study on Action Space

Inspired by the Frame Skip [3] trick which restricts the agent to only observe the environment and act once every $k$ frames rather than one frame, [11] propose to use action bundle to let the agent predict $k$ strokes at each step and the renderer renders these strokes in order. Fix the total number of strokes $N$ in a painting process, [11] reports that $k = 5$ is their best hyperparameter. As we find that $k = 1$ is hard to converge, we experiment with $k = 3$ and $k = 5$ for comparison. The qualitative analysis in Figure 6 indicates that, the agent with $k = 3$ behaves more aggressively compared with $k = 5$ at each step, sometimes yields unsatisfactory control over output actions such as inaccurate color estimates.

---

[2]Specifically, $C_t, I_t \in R^{3 \times H \times W}$ are three channels images, $P \in R^{2 \times H \times W}$ and represent $x$ and $y$ coordinate information with separate channels. The timestamp scalar $T$ is expand to all pixels as $T \in R^{1 \times H \times W}$.

Note that $k = 1$ is regular stroke-by-stroke painting process, while $k = N$ reduce the problem into a one-pass parameter searching as in [28]. Letting $k = 5$ can indeed strike a trade-off balance between exploration on action combinations and the computational expenses.

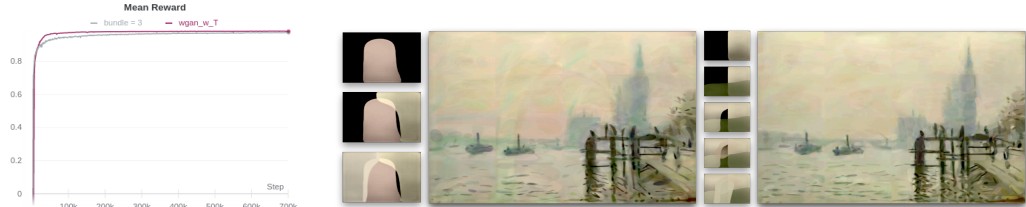

Figure 6: Comparison with action bundle $k = 3$ and $k = 5$ (wgan_w_T in the lengend).

## 4.3 Study on Reward Design

Selecting a suitable metric to measure the difference between the current canvas and the target image is found to be crucial for training a painting agent. The reward $r(s_t, a_t) = L_t - L_{t+1}$ encourages the agent to take action that can bring more closeness to the target image compared on last step. [11] uses WGAN [1] reward, motivated by the widely used discriminator loss in GAN field in generating fine-grained details. Formally, the WGAN reward is obtained by $D(C_{t+1}, I) - D(C_t, I)$. Unlike in normal GAN-based image generation where the Discriminator takes single image as input, here we construct a paired tuple $(C_t, I)$ to the Discriminator, in the hope that it can extract even nuances between $C_t, I$ when the pixel-wise similarity approaching saturated. Compared with fixed $L_1$ and $L_2$ reward, WGAN has the potential to accurately learn an appropriate reward function for this specific task. We denote $(I, I)$ as real samples and $(C_t, I)$ as fake samples. We find that using $L_1$ and $L_2$ rewards can also perform relative satisfactory results compared to GAN reward at the sacrifice of certain details, which is in correspondence with our expectations.

Another line of approach that we experimented is to replace the above mentioned pixel-level loss to perceptual losses, which are widely used in image generation in the computer vision area. A natural extension we came up was to explore image style transfer. For example, given a photo of girl, a human painter is able to draw the portrait sketch by changing the style of strokes while keep the content similar to the original one. We modify the state representation by taking both content image and style images, and calculate the reward at each step by calculating their style/content losses respectively. We use pretrained VGG network and corresponding layers as suggested in [12]. However, we find it really hard to train the painter with pure perceptual losses. We therefore take a step back by replacing the GAN reward with VGG reward only, and find that the agent behave similar with that in SPIRAL [5]. We conjecture that a strong and direct reward signal from pixel level comparison is critical to such painting environment. (See full comparison between different reward design in Figure 7.)

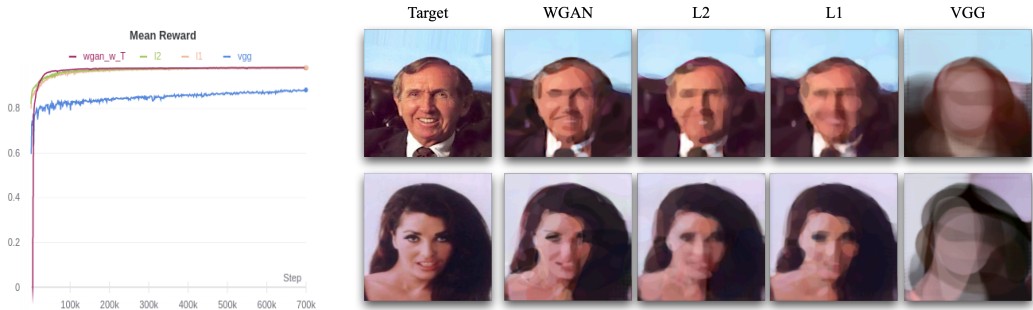

Figure 7: Comparison on different reward design. Overall, L1/L2/GAN reward have similar good performance, while VGG reward can not give enough strong signal to bring photo-realistic lookings.

### 4.4  Study on Algorithms

**Deterministic Policy Graident**    Since the action space in stroke geneartion are mainly continuous and high dimensional, discretizing the action space to adapt to discrete DRL methods such as DQN [18] and PG [23] is quite burdensome. DPG [20] uses deterministic policy to resolve the difficulties caused by high-dimensional continuous action space. Learn2Paint [11] adopts a model-based Deep Deterministic Policy Gradient (DDPG) [14] method to train the agent, which is suitable for high dimensional continuous action spaces. Learn2Paint uses four neural networks: a Q network, a deterministic policy network, a target Q network, and a target policy network. The actor models a policy $\pi$ that maps a state $s_t$ to action $a_t$. The critic estimates the expected reward for the agent taking action $a_t$ at state $s_t$. We note that DDPG works well in most settings in our task. Additionally, we also implement TD3 [4] and SAC [8] to see whether the proposed modifications will bring improvements in this task. It turns out that, the 1) Clipped Double-Q Learning, the 2) Delayed Policy Update, and the 3) Target Policy Smoothing tricks bring no significant improvements. Meanwhile, the introduced entropy regularization in SAC will deteriorate the original performance. We conclude that in such high dimensional continuous action spaces, deterministic policies are still preferred. A notable disadvantage of pure deterministic policy here is that, in early training period, the agent may fall into deadlock by repeatedly cover its previous strokes. Therefore, we introduce certain randomness by adding gaussian noises with $N(0, \sigma)$, where a small $\sigma \geq 0$ denotes the noise level. We find that the priginal training scheme remains robust with the presence of such noises, and greatly reduce the probability of strokes repeating themselves.

### 4.5  Study on Training Datasets

At first we thought that the agent is agnostic to the type of dataset, as the task of copy painting is purely pixel value based. We use CelebA [17] as our default training data. When testing the trained model on other type of images, such as landscape paintings where large strokes are always expected, we find that line-shape strokes will always appear at early stages. Investigating the learned policy of our trained agent on CelebA, we find that when the timestamp is taken in state representations, our agent has learned to paint the parts of human eyes and eyebrows with straight line strokes. This implies a side-effect of training with timestamp indicator in the way that the painting agent has the tendency possibility to "remember" commonly appeared strokes in the training dataset even under complete different scenes. Similar phenomenon is also observed when we change dataset to OmniArt dataset [22] where the majority of training images are very complicated, with many fine strokes or slim lines. Meanwhile, many paintings in old times are painted on supports such as wooden panels, where the texture of materials are randomly spread across the whole paintings. Such patterns will also be inherited when directly apply the trained model to other type of images.

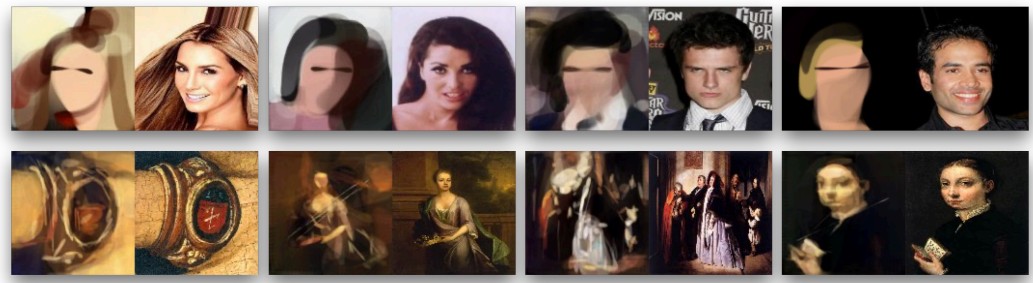

Figure 8: Illustration of 1) the straight-line strokes around eyes in CeleA dataset, and 2) the slim-line artifacts in OmniArt dataset.

## 5  Discussions

The above introduced RL painting agent can already achieve very good performance in many simple scene scenarios. This drives many people to further explore this stroke-based "Learning to Paint" tasks and discuss about current limitations. There is a number of newly published papers have started explorations. **Improve Granularity:** Noting that the existing methods struggle to cover the

granularity and diversity possessed by real world images, [21] propose a semantic guidance pipeline to distinct between foreground and background brush strokes as $a_t = \{a_b, a_f\}$, and introduce both the zoom-in trick and in-focus object detection into the system; **Style Transfer** Stylized Neural Painting [28] re-frame the stroke prediction as a purely parameter searching processing, and thus can be naturally fits the neural style transfer framework; **Optimal Order:** Pixelor [2] present a competitive drawing gam scenarios, where the participant whose sketch is recognized first is a winner. They infer the stroke order that maximizes early recognizability of human training sketches. Then use these orders to supervise the training of a sequence-to-sequence stroke generator; **Generalized Design:** Inspired from the success of Learn2Paint, PlotThread [24] develop a reinforcement learning framework that trains an AI agent to design storylines which is also a sequntial line generation process; **Time Lapse:** TimeCraft [26] proposed a recurrent probabilistic model that captures the stochastic decisions of human artists. They view the process as a video synthesis task which synthesize time lapse videos by learning from real painting time lapse data. One future direction we curious about is to bridge the RL painter and GAN based image generation to bring in the generative ability for the agent, where the inconsistency between state observation and the reward signals still leaves room for study.

## 6   Conclusion

Inspired by Learn2Paint [11], the idea of building a RL agent which is capable of dissecting arbitrary images into step-by-step strokes intrigues us. We conduct detailed experiments with alternatives on each module designs, with the aim to demystify how it actually works and explore the potential for developing more flexible painting agents. Our experiments reveal several key components which play critical roles in training a successful painting agent with accurate strokes, which were not mentioned in the original paper and also overlooked by us. Specifally, We conclude that 1) a well trained neural renderder, the 2) inclusion of timestamp $T$ as an indicator of action affordance, 3) the combined strokes in each step in the format of action bundle, and the provided 4) pixel-level reward signals are crucial for training the painting agent with our model-based DDPG RL algorithms. We also point out certain restrictions caused by the stroke design and dataset bias. In this way, a variety of modification can be made by replacing traditional strokes and changing different datasets, which allows us to develop more natural-looking painting agents in tailored applications.

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
