# OpenReview forum: "Demystify Painting with RL"
_CUHK.edu.hk/2021/Course/IERG5350_

### Official Review · AnonReviewer2 · 2020-12-19
**Good research about effect of different modules in painting with RL**

**Rating:** 8
**Confidence:** 3

**Review:**

Overview:
The paper explore and research the effect of different module designs for learn2Paint by experiments with alternatives on each module designs, including state representation, action space, reward design, algorithms and training database.

Pros:
I think this paper has clear logic and detailed content. Its experiments requires a lot of work.It tries the different designs of several modules and compares their performance.It is meaningful to better use DDPG for painting more accurately.

Cons:
In each module, it just apply different current tricks or change the current parameters to compare the effect of them. It does not have its own novelty. Although I think the work is enough good, if it can  explore more will be perfect.
In 4.4, there are not reasons why it gets conclusion that Clipped Double-Q Learning, the Delayed Policy Update, and the Target Policy Smoothing tricks bring no significant improvements. It will be more convinced by adding the experiment records.

---

### Official Review · AnonReviewer3 · 2020-12-20
**This article builds their RL agent to dissect arbitrary images into step-by-step strokes with detailed analysis.**

**Rating:** 8
**Confidence:** 4

**Review:**

General:

Significance: This article buids a RL agent to dissect arbitrary images into step-by-step strokes.

Novelty: The work shows their innovation on building RL agent.

Technical quality: This paper conducts experiments under different algorithmic designs to dissect images with detailed analysis.

Clarity: The clarity is good and easy to follow. This paper defines the problem clearly and uses pictures to help readers better understand it.

Specific:

Pros: a. Define the problem of demystify painting; b. Build their own RL agent; c. Show detailed analysis.

Cons: a. It could be better for readers to understand the efficiency of your own agent if you could use a baseline model to compare with your RL agent and mention it at the conclusion.